# Zoomer: **Adaptive Visual Token Allocation for Black-box MLLM**

## Abstract

Multimodal large language models (MLLMs) such as GPT-4o, Gemini Pro, and Claude 3.5 have enabled unified reasoning over text and visual inputs, yet they often hallucinate in real-world scenarios—especially when small objects or fine spatial context are involved. We pinpoint two core causes of this failure: the absence of region-adaptive attention and inflexible token budgets that force uniform downsampling, leading to critical information loss. To overcome these limitations, we introduce Zoomer, a visual prompting framework that delivers token-efficient, detail-preserving image representations for black-box MLLMs. Zoomer integrates (1) a prompt-aware emphasis module to highlight semantically relevant regions, (2) a spatial-preserving orchestration schema to maintain object relationships, and (3) a budget-aware strategy to adaptively allocate tokens between global context and local details. Extensive experiments on nine benchmarks and three commercial MLLMs demonstrate that Zoomer boosts accuracy by up to 27% while cutting image token usage by up to 67%. Our approach establishes a principled methodology for robust, resource-aware multimodal understanding in settings where model internals are inaccessible.

## 1 Introduction

Vision-language understanding has emerged as a central challenge in multimodal artificial intelligence, with broad applications ranging from robotics and autonomous driving to scientific diagram analysis and human-computer interaction. Recent advances in multimodal large language models (MLLMs), such as GPT-4o, Gemini Pro, and Claude 3.5, have significantly pushed the boundaries of this field by enabling unified reasoning over text and visual inputs Li et al. (2024); Gu et al. (2024). These models have demonstrated impressive performance on a range of benchmarks and are increasingly perceived as possessing near-human or even PhD-level capabilities in controlled environments. However, despite these promising results, our large-scale empirical study uncovers a critical limitation that remains underexplored: MLLMs exhibit a pronounced hallucination tendency in real-world visual reasoning tasks, largely due to their inherent difficulty in perceiving small objects and limited field-of-view awareness. This "small object blindness" can lead to severe and often undetected reasoning failures, raising concerns about the deployment of MLLMs in real-world visual reasoning scenarios.

To investigate the root causes of hallucination in MLLMs, we conduct a systematic analysis and uncover two fundamental limitations in the visual processing pipeline of current black-box MLLMs. First, these models lack region-adaptive attention mechanisms, resulting in uniform spatial processing that disregards task-relevant visual saliency. Despite their architectural complexity, black-box MLLMs fail to emulate the human visual system's ability to focus selectively while preserving contextual structure. As illustrated in Figure 1, this limitation leads to persistent failure even in elementary tasks such as object counting. Notably, the issue is consistent across different model families and prompting strategies, indicating a fundamental limitation rooted in the design of their visual tokenization and attention mechanisms, rather than implementation-specific flaws.

Second, we identify a critical tension between token budget constraints and visual fidelity. Black-box MLLMs are constrained by fixed token limits for both textual and visual inputs—an engineering choice originally motivated by inference efficiency and fairness considerations (Chen et al., 2024). However, our analysis reveals

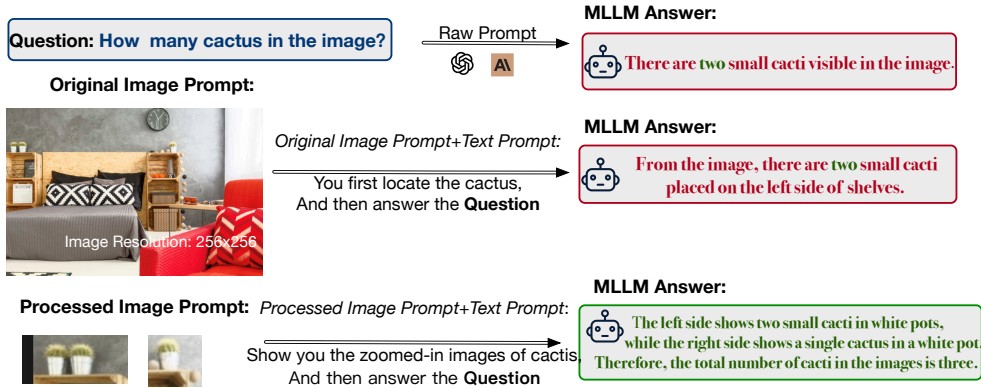

Figure 1: Illustration of a black-box MLLM's approach to counting cacti in an image. The model identifies two small cacti on the left side and overlooks the single cactus on the right side of the image, arriving at a total of three cacti. The processed prompt highlights specific regions of interest to facilitate the correct object count.

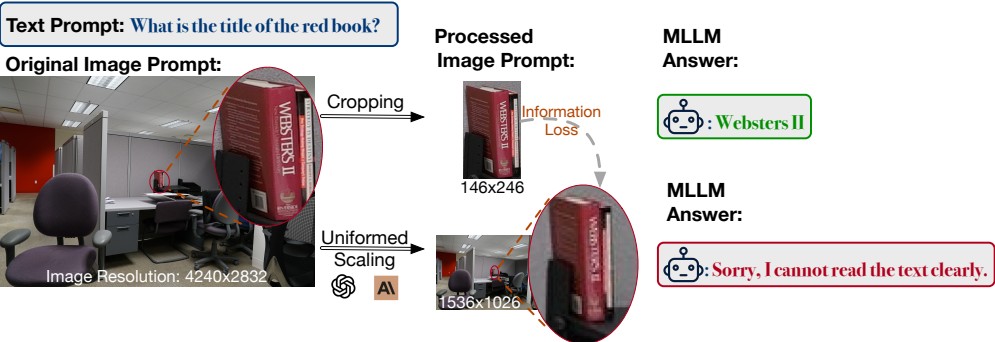

Figure 2: Illustration of information loss during image processing in black-box MLLMs. The original high-resolution image (4240x2832) is downscaled to meet token limits (1536x1026), leading to the loss of critical details. Cropping to focus on a region of interest (146x246) allows the model to correctly identify the book title as "Webster's II".

that token budgeting in vision tasks is a core computational bottleneck that directly impacts perception granularity. As shown in Figure 2, the commonly adopted strategy of uniform image downsampling leads to substantial loss of local detail, especially for small or spatially distributed objects. Crucially, we demonstrate that neither prompt engineering nor improved resizing algorithms can compensate for this representational loss.

These observations motivate us to formally define a new problem in visual prompting for MLLMs: **How can region-aware, spatially coherent visual inputs be constructed under strict token constraints in black-box inference settings?** We argue that solving this problem requires a paradigm shift from heuristic prompt design to principled visual token orchestration. We decompose this problem into three interdependent sub-tasks: (1) **Region Selection** – identifying and prioritizing task-relevant regions without model internals; (2) **Spatial Preservation** – retaining spatial relations and object continuity across selected regions; and (3) **Budget-Aware Prompting** – balancing the allocation of tokens between global context and fine-grained details.

To address these challenges, we present Zoomer, a visual prompting framework designed for token-efficient, region-aware visual representation. Zoomer comprises three integrated components:

- **A prompt-aware visual emphasis module**, which selectively highlights salient regions by manipulating image input structure, enabling focused attention without altering the model;
- **A spatial-preserving orchestration scheme**, which ensures that contextual coherence and relative object positioning are preserved across split or patched inputs;

- **A budget-aware region selection strategy**, which dynamically allocates token capacity based on content density and task relevance.

We evaluate Zoomer on a suite of diverse benchmarks, including Vstar(Wu & Xie, 2023), CVBench(Tong et al., 2024a), and RealWorldQA (xAI, 2024), which collectively capture controlled, open-domain, and in-the-wild reasoning scenarios. Zoomer consistently outperforms competitive baselines in both accuracy and efficiency. On the Vstar dataset, the Zoomer-Patches variant achieves a 26.9% absolute accuracy gain, while on RealWorldQA, Zoomer-Adaptive exceeds baseline performance by 12.1%. Moreover, in the TerraIncognita setting, our method reduces token usage by 67% while improving accuracy by 6.4%. These gains hold across multiple API providers, including GPT-4o[1], Gemini-1.5Pro[2], and Claude-3.5-Sonnet[3], demonstrating that Zoomer generalizes well across model architectures and tokenization schemes.

In summary, this work makes the following contributions: (1) We provide the first in-depth empirical analysis of how token allocation and spatial attention impact visual reasoning performance in black-box MLLMs. (2) We formulate the token-constrained, region-aware visual prompting problem and identify its core design dimensions. (3) We propose Zoomer, a general-purpose framework for token-efficient, spatially coherent visual prompting. (4) We demonstrate that Zoomer improves both accuracy and efficiency across models, datasets, and domains, highlighting a new path toward robust, resource-aware multimodal understanding.

## 2 Pilot Experiments

A fundamental challenge in black-box multimodal large language models (MLLMs) is their inability to process high-resolution visual inputs efficiently under strict token budget constraints. In platforms such as GPT-4o, visual inputs are internally segmented into fixed-size image patches (typically 512×512 pixels), each corresponding to a predefined token cost (e.g., 170 tokens per patch)[4]. This patch-based tokenization strategy, designed for balancing inference efficiency and fairness, introduces a critical trade-off: higher image resolution yields more detailed information but leads to rapid token consumption, limiting the number of patches a model can process. Conversely, reducing token usage through downsampling or cropping risks omitting essential fine-grained visual cues.

To quantify the impact of this trade-off, we conducted a series of pilot experiments using GPT-4o-0513 on the *Vstar-Bench* dataset, which requires precise visual grounding of small or occluded objects in complex scenes. We compare three visual prompting strategies: (1) **Unaltered Input**, where the original image is submitted directly without preprocessing; (2) **Image Crop**, Unless otherwise stated, all regions of interest (ROIs) used in this paper are obtained automatically at inference time using the multi-scale, prompt-aware open-vocabulary detector pipeline described in Section **??**. We do *not* use any ground-truth bounding boxes from the datasets when constructing ROIs, including the introductory examples in this section; and (3) **Zoomed Crop**, which magnifies the cropped region to maximize visual feature visibility while preserving the 512×512 input constraint. Each cropped ROI is normalized to a consistent visual scale before composition. Concretely, we resize each ROI such that its longer side equals 512 pixels while preserving aspect ratio. ROIs that are originally smaller than this scale are not up-scaled beyond their native resolution, in order to avoid excessive blurring or hallucinations. For large input images, we first crop ROIs using the detector and then apply this max-side–512 rule per ROI. This normalization ensures a consistent scale before spatial composition in Section 4.2.

The results in Table 1 reveal several critical insights. While **Unaltered Input** provides full-scene context, it incurs excessive token cost (955 tokens), with limited performance gains—suggesting that high token usage alone does not guarantee better model understanding. In contrast, the **Image Crop** method reduces token usage substantially but fails to improve accuracy, likely due to the lack of visual emphasis and limited discriminative detail in the cropped region.

---

[1]https://platform.openai.com/
[2]https://gemini.google.com/
[3]https://anthropic.com/
[4]https://openai.com/api/pricing/

| Method | Accuracy | Prompt Tokens |
|---|---|---|
| Unaltered Input | 57% | 955 |
| Image Crop | 58% | 270 |
| Zoomed Crop | 64% | 270 |

Table 1: Performance of different methods on Image from Vstar.

The **Zoomed Crop** strategy significantly outperforms both baselines, achieving 64% accuracy with only 270 tokens. This improvement highlights the importance of visual emphasis: by magnifying relevant regions, **Zoomed Crop** restores critical visual granularity lost in standard downsampling, while maintaining input compactness. Importantly, this approach respects the model's fixed patch budget without requiring architectural changes or access to internal weights.

These findings underscore a fundamental limitation in black-box MLLMs: existing visual prompting techniques fail to manage the resolution–token trade-off effectively. Naïve approaches such as downscaling or region cropping without adaptive enhancement do not sufficiently preserve semantic or structural information. Our pilot experiments suggest that vision enhancement strategies—particularly those that incorporate selective magnification—are essential for preserving task-relevant detail in constrained inference settings.

In subsequent sections, we build upon this insight to design Zoomer, a visual prompting framework that generalizes the principles of adaptive emphasis, spatial preservation, and budget-aware token allocation across diverse visual reasoning tasks.

## 3 Related Work

### 3.1 Multimodal LLMs: Open-Source and Black-Box Models

The integration of visual and textual modalities in large language models (LLMs) has led to significant advancements in multimodal models (MLLMs) like GPT-4o, Gemini Pro and Claude3-Sonnet. These models rely on effective visual encoding strategies to bridge the gap between language and vision. Approaches such as CLIP (Yang et al.) align visual and language embeddings through contrastive learning, while models like Flamingo (Alayrac et al.) and BLIP-2 (Dai et al.) use cross-attention mechanisms or pretraining modules to link vision encoders with LLMs. However, these methods often rely on fixed low-resolution inputs (e.g., 224x224), limiting their ability to process high-resolution images or non-standard aspect ratios (Liu et al., a), which hampers performance on fine-grained tasks such as OCR and small object detection

In contrast, open-source multimodal models (Li et al., c; Xu et al.; Zhang et al., a; Li et al., a; Zhao et al.) allow for architectural modifications and fine-tuning to accommodate any-resolution inputs. However, black-box MLLMs such as GPT-4o and Gemini Pro, which impose strict token limits for computational efficiency, require alternative solutions. The need to downsample or crop images to meet these constraints often results in the loss of crucial visual details, particularly in tasks requiring detailed visual understanding. While position embedding interpolation (Bai et al.; Wang et al.; Luo et al.; Hong et al.; Chen et al.) and patch-based cropping (Xu et al.; Li et al., a) widely adpoted in open-soure models offer promising directions for any aspect ratio and any-resolution image processing, they are not applicable to black-box models, where architectural changes and extra training/fine-tuning are not permitted.

### 3.2 Object Detection

Traditional object detection models, such as Faster R-CNN (Ren et al.) and YOLO (Redmon et al.), effectively identify and localize objects within predefined categories. However, they struggle with open-set scenarios, where novel objects not seen during training need to be detected.

Recent advances address this limitation through open-set detection models that leverage natural language processing. For instance, OV-DETR (Zang et al.) integrates CLIP with object detection to generate

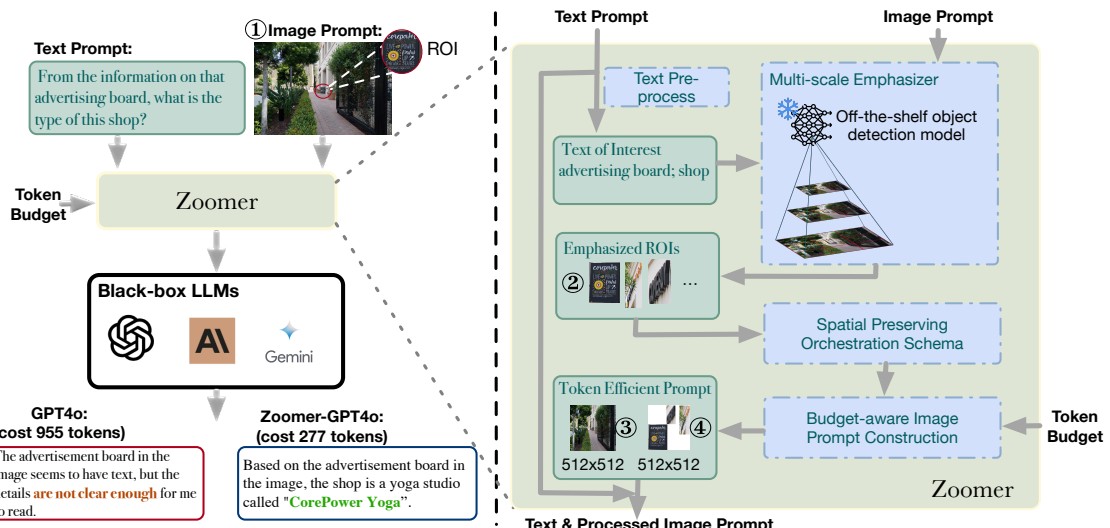

Figure 3: The Zoomer framework. Left: Raw Input image (①) and text prompt are processed by Zoomer and then fed into a black-box LLM (e.g., GPT-4o) for analysis, resulting in more accurate and detailed responses compared to standard input methods with even token saving. Right: Zoomer processes the text to extract key terms and uses a multi-scale emphasizer(§4.1) with an off-the-shelf object detection model to identify regions of interest (ROIs). The identified ROIs (②) are then processed through a spatial preserving orchestration schema (§4.2) for a filtered emphasized patch (④) and a budget-aware image prompt construction module (§4.3) to create a token-efficient prompt within the specified budget. A scaled global view (③) is also generated for potential prompting.

category-specific bounding boxes from textual prompts, enabling detection in open-world settings. Similarly, GLIP (Li et al., b) reframes detection as a grounding problem, improving alignment between visual regions and textual descriptions. DetCLIP (Yao et al.) extends this further using pseudo labels from large-scale captioning datasets, enhancing generalization. Grounding DINO (Liu et al., b), built on the DETR framework (Carion et al.), also advances open-set detection through natural language integration.

In addition, SAM (Kirillov et al., 2023) and SAM-2 (Ravi et al., 2024) offer zero-prompt or minimal-prompt segmentation for arbitrary objects but lack robust text-prompt handling. EVF-SAM (Zhang et al., b) overcomes this by extending SAM's capabilities to better manage complex text-based object segmentation.

By incorporating these models, Zoomer enhances its ability to dynamically detect and emphasize regions of interest (RoIs), enabling black-box MLLMs to focus on the most relevant visual content without losing critical details, which is essential for maintaining high performance across varied resolutions.

## 4 Method Overview

Building upon the findings of our pilot experiments, we introduce Zoomer, a unified visual prompting framework that enables detail-preserving, token-efficient encoding of high-resolution visual inputs for black-box multimodal LLMs. Existing models such as GPT-4o and Gemini 1.5 often rely on uniform downsampling or tiling to fit image inputs into fixed-size patches, which leads to significant information loss and degraded performance on fine-grained vision tasks.

Zoomer addresses this challenge through three tightly coupled modules, as shown in Figure 3:

- **Prompt-Aware Visual Emphasizer**: extracts task-relevant image regions based on semantic cues derived from natural language prompts.
- **Spatial-Preserving Orchestration Schema**: reconstructs extracted regions into spatially faithful layouts that preserve object relationships and scene structure.
- **Budget-Aware Prompting Strategy**: manages token allocation under user-specified constraints by adapting the number and organization of image regions.

### 4.1 Prompt-aware Visual Emphasizer

The prompt-aware visual emphasizer utilizes a multi-scale emphasizing strategy to prioritize image slices that are most relevant to the input prompts. By analyzing the semantic content of the prompts, this component dynamically selects and enhances specific regions of the image at varying resolutions. This approach not only enriches the contextual information available to the model but also mitigates the adverse effects of losing critical details during the resizing process.

**Prompt Tokenization** Prompt tokenization is a critical first step in which input prompts are parsed into meaningful tokens. This process segments the prompt into components that can be easily analyzed for semantic relevance. Specifically, the prompt is divided into structural components, and our focus is on processing the relevant sections that contribute directly to visual emphasis.

To enhance the extraction of semantically relevant tokens, we apply advanced natural language processing (NLP) techniques. First, we use the NLTK library[5] to remove stopwords, reducing noise and ensuring that the model's attention remains on the most critical visual elements. By eliminating these non-essential words, we concentrate on key terms that directly influence the visual emphasis.

In addition to basic stopword removal, we utilize dependency parsing (Sarthi et al., 2024; De Marneffe & Manning, 2008) to analyze the syntactic structure of the prompt. We use a lightweight dependency parser to identify noun phrases and their syntactic roles (e.g., question objects versus modifiers). Rather than constructing a full scene graph, we use these dependencies as an *importance-ranking* mechanism: (1) higher-priority noun phrases are passed to the open-vocabulary detector first, and (2) when multiple entities compete for limited detection budget, the ranking is used to break ties. In other words, dependency parsing is used for importance weighting and ordering prior to detection, not for explicit relation modeling. This deeper analysis identifies core entities and relationships, such as subject-object pairs and action verbs, which are crucial for interpreting the user's intent. By focusing on these core semantic elements, we ensure that the visual emphasis aligns precisely with the underlying meaning of the prompt.

Finally, we strip away any irrelevant formatting or non-content-related details, allowing the visual emphatizer to focus solely on the essential information. This multi-layered tokenization approach ensures an optimal match between the tokenized prompt and the image features selected for emphasis.

**Multi-Scale Emphasizing Algorithm:** Given a key object term extracted from the text prompt, the Multi-Scale Emphasizing Algorithm 1 utilizes a state-of-the-art object detection model to localize the corresponding object in the image prompt. In our experiments, we primarily employ GroundingDINO (Liu et al., b) as our localization model.

The encoder in such models typically downsamples the input image to a resolution of $224 \times 224$ or $336 \times 336$, potentially resulting in information loss when localizing the target object at a coarse granularity. To address this limitation, we propose a Multi-Scale Emphasizing Algorithm that processes the original image at multiple resolutions. The algorithm divides the input image into patches at various granularities, *e.g.,* $2 \times 2$, $3 \times 3$, and beyond. For each generated patch, we apply the object detection model to localize the target object. The algorithm retains bounding boxes returned by the model that exceed a predefined confidence threshold. These high-confidence bounding boxes collectively form the output of our algorithm, providing a comprehensive multi-scale representation of the target object's location.

### 4.2 Spatial-preserving Orchestration Schema

Building upon the Multi-Scale Emphasizing Algorithm, we introduce a Spatial-preserving Orchestration Schema to maintain the structural integrity of the image during the encoding process. This schema filters the bounding boxes obtained from the Multi-Scale Emphasizing Algorithm and ensures that the relative positions of the selected image slices are preserved, facilitating a more faithful representation of the original image layout and enabling coherent reconstruction when processed by the multimodal LLM. To refine the selection of bounding boxes, we implement a Non-Maximum Suppression (NMS) based slice filtering method. NMS is employed to eliminate redundant and overlapping slices, retaining only the most salient features that align with the prompt. The process works as described in Algorithm 2.

---

[5]https://www.nltk.org/

---

**Algorithm 1** Multi-Scale Emphasizing Algorithm

---

**Require:** $I$: input image, $k$: key object term, $M$: object detection model, $T$: confidence threshold
**Ensure:** $B$: set of bounding boxes
 1: $B \leftarrow \emptyset$
 2: $S \leftarrow \{2, 3, \ldots, S_{\max}\}$               $\triangleright$ Set of scaling factors
 3: **for** each $s \in S$ **do**
 4:      $P_s \leftarrow \texttt{DivideIntoPatches}(I, s \times s)$
 5:      **for** each patch $p \in P_s$ **do**
 6:          $b, c \leftarrow M(p, k)$              $\triangleright$ Get bounding box and confidence
 7:          **if** $c \geq T$ **then**
 8:              $B \leftarrow B \cup \{b\}$
 9:          **end if**
10:      **end for**
11: **end for**
12: **return** $B$

---

By setting an appropriate threshold $T$ for the Intersection of Union (IoU) of bounding boxes around the selected regions, we ensure that only the highest-quality slices are retained for the encoding process. This filtering step enhances computational efficiency by reducing the number of slices to be processed and improves the clarity and relevance of the visual information provided to subsequent stages of the model.

The resulting set of filtered slices are then orchestrated to preserve their original relative positions within the image. This orchestration process involves the following steps: **Slice Extraction**: For each bounding box $b_i$ in the filtered set $F$, we extract the corresponding image slice from the original image. **Blank Image Creation**: We create a new blank image with the same dimensions as the original image. **Slice Placement**: We place each extracted slice onto the blank image at its original position, leaving the rest of the image blank. **Image Shrinking**: The resulting image, containing only the selected slices in their original positions with the rest left blank, is then shrunk to a predetermined size while maintaining its aspect ratio.

In all configurations, Zoomer composes the selected ROIs and optional global view into a *single* image canvas, which is then passed to the black-box MLLM. In particular, in Zoomer-Adaptive and Zoomer-Global, ROIs are placed according to their original relative coordinates using the spatial-preserving layout, and, when enabled, a down-scaled global overview is appended as an additional strip on the same canvas. Thus, the MLLM always receives exactly one reconstructed image rather than multiple separate images.

---

**Algorithm 2** NMS-based Slice Filtering

---

**Require:** $B$: set of bounding boxes, $T$: IoU threshold
**Ensure:** $F$: set of filtered bounding boxes
 1: $F \leftarrow \emptyset$
 2: Sort $B$ in descending order of confidence scores
 3: **while** $B \neq \emptyset$ **do**
 4:      $b_{\max} \leftarrow \arg\max_{b \in B} \text{score}(b)$
 5:      $F \leftarrow F \cup b_{\max}$
 6:      $B \leftarrow B \setminus b_{\max}$
 7:      **for** each $b \in B$ **do**
 8:          **if** $\text{IoU}(b_{\max}, b) \geq T$ **then**
 9:              $B \leftarrow B \setminus b$
10:          **end if**
11:      **end for**
12: **end while**
13: **return** $F$

---

### 4.3 Budget-aware Prompting Strategy:

Our approach incorporates a sophisticated budget-aware prompting strategy that optimizes the allocation of token budget for image processing. This strategy begins with a user-specified total token budget $B_{\text{total}}$, allowing for customization based on specific task requirements or computational constraints. We propose four varieties of Zoomer to accommodate different budget scenarios and task requirements:

- **Zoomer-Local(④):** This variant utilizes only the spatial-preserving schema to consolidate all focused image slices into a single image patch(④ in Figure 3). It is optimal for scenarios with very limited token budgets, prioritizing the most relevant visual information.
- **Zoomer-Adaptive (④ + ◇ ③ ):** This approach dynamically includes a global view of the original image if the cropped portion falls below a certain threshold $T_A$. This allows the MLLM to better understand the overall scene context when the budget permits, while still focusing on key areas of interest.
- **Zoomer-Global (④ + ③):** This variant assigns a global view to all images, regardless of the specific regions of interest. It is suitable for tasks that require consistent overall context and when the token budget is sufficient to include both global and local information.
- **Zoomer-Patches(② + ③):** This is the most token-intensive approach, assigning each image slice its own patch without spatial preservation, along with a global view. It provides the most detailed information but requires the largest token budget.

**Adaptive global view decision.**

$$T_A = \frac{\text{total area of all cropped ROIs}}{\text{area of the original image}} \tag{1}$$

Let EQ 1 denote the fraction of the original image covered by the selected ROIs. Zoomer-Adaptive decides whether to append a global overview based on $T_A$: if $T_A \leq 0.35$, we append a down-scaled global view to preserve overall scene context; if $T_A > 0.35$, we omit the global view to save visual tokens, since the ROIs already cover a sufficiently large portion of the scene. The threshold 0.35 is chosen empirically on the VSTAR validation set to balance accuracy and token efficiency.

The selection among these varieties depends on the user-specified budget and the nature of the task. For each variant, the number of high-resolution slices or patches $N$ is calculated based on the available budget and the token cost per slice or patch. These slices are selected from the output of our Multi-Scale Emphasizing Algorithm, prioritizing based on their relevance to the key term of text prompts. To present the methods more clearly and vividly, we refer to Figure 4, which outlines the methodology, and Figure 5, which showcases a specific case study.

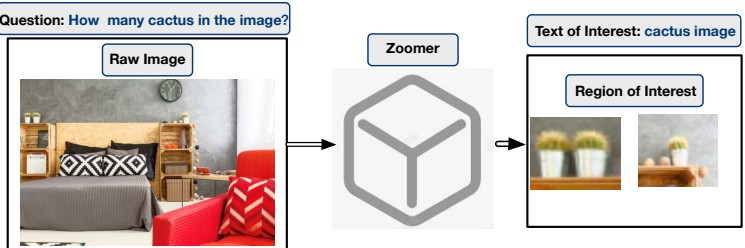

Figure 4: The example of applying Zoomer

## 5 Experiments

We evaluate the performance of Zoomer on a diverse set of visual reasoning tasks using multiple black-box MLLMs. Our experiments are designed to assess both quantitative gains in accuracy and qualitative improvements in visual fidelity under constrained token budgets.

Specifically, our evaluation is guided by the following research questions: **RQ1: Accuracy.** Does Zoomer improve task accuracy across different black-box MLLMs on image-grounded reasoning tasks, compared

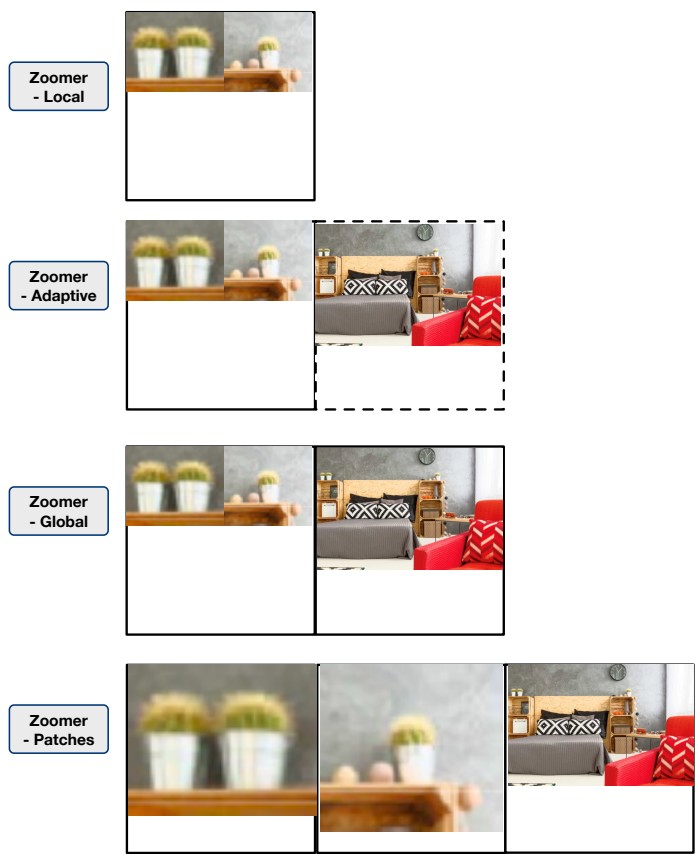

Figure 5: The example of different settings of Zoomer

| Acc./Tokens Bench Method | Vstar | CVBench-2D | CVBench-3D | RealworldQA | SQA-I | MMVP | MMMU | HR-4K | HR-8K |
|---|---|---|---|---|---|---|---|---|---|
| Raw | 56.5%/955 | 68.5%/428 | 78.2%/895 | 67.6%/998 | 87.3%/353 | 83.3% /270 | 68.4%/608 | 50.6%/1105 | 46.8%/1105 |
| Resize | 41.9%/270 | 66.3%/270 | 75.2%/270 | 61.1%/270 | 86.8%/270 | 83.3%/270 | 62.9%/270 | 35.8%/270 | 33.4%/270 |
| Zoomer-Local | 67.1%/270 | 72.4%/270 | 86.2%/270 | 72.4%/270 | 88.3%/270 | 87.1%/270 | 59.8%/270 | 58.8%/270 | 57.7%/270 |
| Zoomer-Adaptive | 67.5%/419 | 72.9%/374 | 87.9%/408 | 74.7%/362 | 91.1%/308 | 88.7%/351 | 61.6%/312 | 60.8%/331 | 59.3%/324 |
| Zoomer-Global | 67.6%/540 | 73.1%/540 | **88.3%**/540 | 75.3%/540 | 92.3%/540 | **88.9%**/540 | 67.3%/540 | **61.3%**/540 | **59.8%**/540 |
| Zoomer-Patches | **71.7%**/1029 | **74.6%**/709 | 85.8%/1113 | **75.8%**/997 | **92.8%**/727 | 88.4%/726 | **68.9%**/841 | 60.4%/713 | 58.9%/875 |

Table 2: Performance of GPT-4o across different datasets using various image prompt processing methods, focusing on accuracy and token consumption. Among these approaches: **Local**: Only the extracted RoIs are used. **Adaptive**: Selectively provides the MLLM with a global view of the image based on the prompt strategy. **Global**: Every request includes the global view of the image. **Patches**: Does not use the Spatial-Preserving Orchestration Schema; instead, each possible RoI is independently provided to the MLLM, including the global view.

to standard prompting baselines? **RQ2: Token Efficiency.** How does Zoomer balance the trade-off between accuracy and token usage, and how does it perform relative to methods that either over-utilize or under-utilize the token budget? **RQ3: Component Effectiveness.** What is the contribution of each major component of Zoomer —specifically, the multi-scale visual emphasis module and the budget-aware prompting strategy—to the overall performance?

**Backbone MLLMs.** We evaluate Zoomer on both commercial APIs (e.g., GPT-4o, Gemini, Claude) and open-source MLLMs. For the latter, we use Qwen2-VL-7B and InternVL2.5-8B as representative models in this work, primarily due to time and resource constraints. As shown in Table 6, applying Zoomer on top of both open-source backbones consistently improves performance across four benchmarks, indicating that

| Method | Accuracy | | Tokens | | Latency | | Money Cost($10-e3) | |
|---|---|---|---|---|---|---|---|---|
| | Zero-Shot | 15-Shot | Zero-Shot | 15-Shot | Zero-Shot | 15-Shot | Zero-Shot | 15-Shot |
| Raw | 78% | 84% | 963 | 13488 | 4.8s | 18.7s | 4.815 | 67.44 |
| Resize | 61% | 74% | 255 | 4080 | 2.9s | 7.5s | 1.275 | 20.4 |
| Low-Detail | 60% | 70% | 85 | 1360 | 2.1s | 6.5s | 0.425 | 6.8 |
| Zoomer-Adaptive | 83% | 88% | 315 | 5112 | 3.1s | 9.8s | 1.575 | 25.56 |

Table 3: Performance in terms of accuracy, latency, and image token cost on TerraIncognita under ICL conditions—specifically with 15 examples per question—and under zero-shot conditions.

| API | Method | *Vstar* | *CVBench-2D* | *RealworldQA* | *MMVP* |
|---|---|---|---|---|---|
| GPT-4o | Raw | 56.5% | 68.5% | 67.6% | 83.3% |
| | Zoomer | 71.7% | 74.6% | 75.8% | 88.9% |
| Gemini-1.5Pro | Raw | 53.1% | 65.4% | 64.0% | 79.8% |
| | Zoomer | 70.4% | 73.2% | 73.9% | 87.8% |
| Claude-3.5-Sonnet | Raw | 51.8% | 66.7% | 61.0% | 80.2% |
| | Zoomer | 69.7% | 72.8% | 74.1% | 87.2% |

Table 4: Accuracy of Different Black-box MLLM APIs. For *Vstar*, *CVBench-2D*, and *RealworldQA*, we used the Patches version of SysName. For *MMVP*, inspired by Table 2, we employed the Global version.

the proposed adaptive visual token allocation is not tied to proprietary APIs. Extending the evaluation to additional high-resolution open-source models such as Qwen3-VL, Qwen2.5-VL, LLaVA-UHD, and MG-LLaVA is an important direction for future work.

## 5.1 Setup

**Assessment and Datasets** We evaluated our system on a series of challenging multimodal tasks, using commercial black-box MLLMs for applications ranging from visual-language reasoning to image understanding and question answering. The experiments were conducted on a variety of different public datasets, including:

1) *Vstar* (Wu & Xie, 2023): A benchmark dataset focused on image classification, used to evaluate fine-grained visual recognition capabilities in object detection and classification tasks.

2) *CVBench* (Tong et al., 2024a): Contains 2 sub-category, $CVBench_{2D}$ and $CVBench_{3D}$, respectively, representing two-dimensional and three-dimensional visual image, respectively, to evaluate the performance of the model when processing images of different dimensions, especially the understanding ability in complex scenes.

3) *RealworldQA* (xAI, 2024): Used to test the multimodal question answering performance of the model in real-world scenarios, involving cross-language and cross-image information processing.

4) *MMVP* (Tong et al., 2024b): A validation set for multimodal visual processing, designed to evaluate the comprehensive understanding of models for complex visual scenes.

5) *ScienceQA* (Lu et al., 2022): A multimodal scientific question-answering dataset featuring multiple-choice questions across a diverse range of science topics.

6) *MMMU* Yue et al. (2024): The validation part of a new benchmark, which designed to evaluate the performance of multimodal models on multidisciplinary tasks that require university-level subject knowledge and deliberate reasoning.

7) *HR* Wang et al. (2024): A high-resolution multimodal benchmark consisting of 4K and 8K images and corresponding questions.

**Models** We employed three black-box MLLMs—GPT-4o-0513, Claude-v3-Sonnet, and Gemini-Pro—accessed via their respective APIs (OpenAI, Claude, Google). Across all experiments, we set the

| Method | Model | Prompt Strategy | *Vstar* | *CVBench-2D* | *CVBench-3D* | *RealworldQA* | *SQA-I* | *MMVP* | *MMMU* |
|---|---|---|---|---|---|---|---|---|---|
| Default | EVF-SAM | Local | 57.1% | 67.6% | 79.9% | 68.5% | 87.8% | 84.0% | 55.3% |
| | | Adaptive | 57.5% | 71.3% | 82.5% | 72.1% | 88.3% | 85.3% | 55.9% |
| | | Global | 57.8% | 72.1% | 83.0% | 72.4% | 88.8% | 87.3% | 56.8% |
| | | Patches | 57.1% | 72.7% | 83.8% | 72.1% | 86.8% | 84.7% | 56.1% |
| | Ground Dino | Local | 58.1% | 70.6% | 82.5% | 71.5% | 85.3% | 84.3% | 55.6% |
| | | Adaptive | 58.3% | 71.5% | 83.9% | 73.1% | 90.1% | 85.6% | 56.3% |
| | | Global | 58.3% | 71.8% | 84.8% | 73.4% | 90.3% | 87.8% | 57.0% |
| | | Patches | 58.8% | 70.6% | 83.1% | 72.6% | 90.9% | 87.7% | 56.6% |
| Multi-Resolution | EVF-SAM | Local | 58.4% | 69.2% | 84.0% | 72.5% | 88.3% | 84.7% | 57.1% |
| | | Adaptive | 58.5% | 71.3% | 85.3% | 73.1% | 91.1% | 86.8% | 58.7% |
| | | Global | 58.4% | 72.1% | 85.8% | 73.4% | 91.8% | 87.8% | 59.6% |
| | | Patches | 60.2% | 71.3% | 85.7% | 73.1% | 91.3% | 86.8% | 59.2% |
| | Ground Dino | Local | 63.6% | 71.8% | 82.6% | 70.2% | 88.8% | 85.1% | 56.8% |
| | | Adaptive | 64.4% | 71.8% | 84.3% | 70.6% | 90.3% | 86.5% | 58.5% |
| | | Global | 66.4% | 72.2% | 84.7% | 71.4% | 91.8% | 86.9% | 59.3% |
| | | Patches | 66.2% | 72.6% | 83.6% | 70.2% | 92.1% | 86.9% | 58.8% |
| Multi-Scale | EVF-SAM | Local | 63.7% | 72.1% | 85.2% | 70.4% | 85.3% | 85.7% | 57.5% |
| | | Adaptive | 64.3% | 72.4% | 86.1% | 70.6% | 90.1% | 87.6% | 58.5% |
| | | Global | 64.3% | 72.8% | 87.9% | 71.7% | 90.3% | 88.8% | 59.9% |
| | | Patches | 67.2% | 73.7% | 87.1% | 73.0% | 90.9% | 88.0% | 59.7% |
| | Ground Dino | Local | 67.1% | 72.4% | 86.2% | 72.4% | 88.3% | 87.1% | 57.7% |
| | | Adaptive | 67.5% | 72.9% | 87.9% | 74.7% | 91.1% | 88.7% | 59.3% |
| | | Global | 67.6% | 73.1% | 88.3% | 75.3% | 92.3% | 88.9% | 59.8% |
| | | Patches | 71.7% | 74.6% | 85.8% | 75.8% | 92.8% | 88.4% | 58.9% |

Table 5: Performance of Zoomer Across Datasets for Different Emphasis Methods, Models, and Prompt Strategies.

temperature to 0 and used greedy decoding for consistency, optimizing the stability of outputs. NMS was applied with a confidence score threshold of 0.8 to filter irrelevant regions from high-resolution images.

**Metrics** We used classification accuracy across all examples as the primary evaluation metric. Additionally, we compared token usage for each model configuration to evaluate the efficiency improvements offered by Zoomer.

**Token efficiency metrics.** Different providers adopt different image tokenization schemes. Whenever an API exposes image-token usage, we directly report the provider's visual token count. When this information is not exposed, we instead count the number of $512 \times 512$ patches that are actually transmitted after reconstruction as a proxy for visual tokens.

To ensure fair cross-model comparison, we always measure token efficiency as a *relative reduction* with respect to each model's own Raw baseline. For a model $m$ and method $x$, we compute

$$\Delta_{\text{tokens}}(m, x) = \frac{\text{tokens}_{\text{Raw}}(m) - \text{tokens}_x(m)}{\text{tokens}_{\text{Raw}}(m)},$$

and pair this with the corresponding accuracy change. This normalization prevents differences in provider-specific tokenization rules from biasing our conclusions about token savings.

**Baselines** We compare Zoomer against the following baseline methods:

1) *Raw*: This baseline feeds MLLM the unmodified prompt, with no adjustments made to the image.

2) *Resize*: Here, images larger than 512x512 pixels are resized to fit within the GPT-4o's patch limit, while smaller images remain unchanged.

### 5.2 Main results

### RQ1: Accuracy Under Token Constraints

We first evaluate the accuracy of Zoomer across multiple visual reasoning benchmarks using GPT-4o. Table 2 compares its performance with baseline prompting strategies, including Raw (unaltered image input), Crop, and heuristic patching. Across all datasets, Zoomer demonstrates consistent and significant gains.

|  | Qwen2-VL-7B | InternVL2.5-8B | Qwen2-VL-7B + Zoomer | InternVL2.5-8B + Zoomer |
|---|---|---|---|---|
| CVBench-2D | 0.648 | 0.656 | 0.683 (+0.035) | 0.697 (+0.041) |
| CVBench-3D | 0.632 | 0.629 | 0.675 (+0.043) | 0.669 (+0.040) |
| RealWorldQA | 0.698 | 0.698 | 0.732 (+0.034) | 0.741 (+0.043) |
| VSTAR | 0.584 | 0.573 | 0.648 (+0.064) | 0.655 (+0.082) |

Table 6: Performance of Qwen2-VL-7B and InternVL2.5-8B with and without Zoomer. Zoomer consistently improves accuracy across all four benchmarks.

| Method | VSTAR | CVBench-2D | CVBench-3D | RealWorldQA |
|---|---|---|---|---|
| Zoomer | 0.717(+0.013) | 0.746(+0.045) | 0.883(+0.090) | 0.758(+0.060) |
| V* | 0.704 | 0.701 | 0.793 | 0.698 |

Table 7: Comparison between Zoomer and a stronger high-resolution baseline V*. V* already uses high-resolution inputs without our cropping and composition.

Notably, Zoomer-Patches achieves 71.7% accuracy on the *Vstar* dataset, outperforming the Raw baseline by 26.9% (from 0.565 to 0.717). On the more challenging *RealWorldQA* dataset, which requires complex visual-linguistic reasoning, Zoomer-Adaptive yields 0.758 accuracy—12.1% higher than the Raw input (0.676). These results indicate that Zoomer enhances the model's ability to ground object references and interpret fine-grained visual cues by preserving high-fidelity regions while respecting token limits.

**RQ2: Cross-Model Generalizability**

To assess generalizability, we apply Zoomer to other closed-source MLLMs, including Claude-3.5-Sonnet and Gemini-1.5Pro. As shown in Table 4, the performance trends hold across architectures. For instance, on *Vstar*, Zoomer improves accuracy from 0.531 to 0.704 on Gemini-1.5Pro—a 32.6% gain. On *RealWorldQA*, Claude-3.5-Sonnet achieves a 34.5% improvement using Zoomer (from 0.610 to 0.741). These results suggest that Zoomer is model-agnostic and portable across different commercial MLLMs without requiring model access or retraining.

As suggested by reviewers, we also evaluate Zoomer on open-source MLLMs. In particular, we use Qwen2-VL-7B and InternVL2.5-8B as representative backbones. As shown in Table 6, applying Zoomer on top of both models yields consistent gains across all four benchmarks (e.g., +0.034–0.043 on RealWorldQA and +0.064–0.082 on VSTAR), confirming that our adaptive image focus mechanism is not tied to proprietary APIs.

**RQ3: Token Efficiency and Latency**

One of the central goals of Zoomer is to maximize information utility under token constraints. Table 3 reports token usage and performance on the TerraIncognita dataset under the ManyICL (Jiang et al., 2024) setting. Zoomer achieves 0.83 accuracy with 315 tokens, while the Raw method reaches only 0.78 with 963 tokens—yielding a 6.4% accuracy improvement with 67% fewer tokens. This substantial reduction in visual token cost translates directly into lower API usage and faster inference.

Moreover, in real-time settings such as autonomous navigation, latency becomes critical. In a zero-shot TerraIncognita setting, Zoomer reduces response time from 4.8 seconds to 3.1 seconds (a 35.4% reduction) without sacrificing accuracy. These results underscore the practical benefits of Zoomer for latency-sensitive and resource-constrained applications such as real-time surveillance, embodied agents, and mobile vision systems.

We further compare Zoomer against a stronger high-resolution baseline V*, which already uses high-resolution visual inputs without our prompt-aware cropping and composition. As shown in Table 7, Zoomer still provides clear gains, especially on CVBench-3D and RealWorldQA. This indicates that adaptive image focus and spatial-preserving orchestration remain beneficial even when the baseline already operates in a high-resolution regime, and strengthens our connection to high-resolution prompting methods such as LLaVA-UHD and MG-LLaVA.

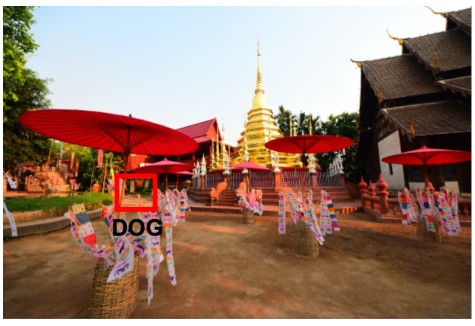 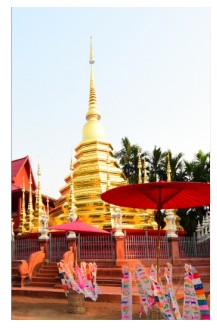 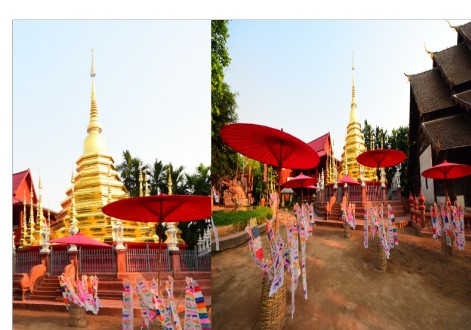

**Raw Image**  **Zoom-Local**  **Zoom- Adaptive**

Figure 6: **Failure case due to missed small object. Left:** raw image (dog highlighted) and the question: *"Is the dog on the left or right side of the golden tower?"* (answer: left). **Middle:** Zoomer-Local reconstruction, where the detector fails to include the dog in any ROI, leading to an incorrect prediction. **Right:** Zoomer-Adaptive reconstruction; appending a global view does not fully resolve the issue when the dog is indistinguishable at the down-scaled resolution.

### 5.3  Findings

While Zoomer-Patches generally performs well under generous token budgets, we observe that in certain datasets it underperforms relative to other variants. For instance, on the $CVBench_{3D}$ dataset, Zoomer-Patches yields an accuracy that is 2.5 percentage points lower than Zoomer-Global and 4.0 points lower than Zoomer-Local. Similarly, on the $MMVP$ benchmark, the Patches variant trails Zoomer-Global by 0.5 percentage points and Zoomer-Adaptive by 0.3 points. These differences persist across multiple trials and are consistent beyond minor fluctuations attributable to model non-determinism.

We hypothesize that this degradation stems from the way Zoomer-Patches encodes visual inputs. Specifically, each RoI is treated as an independent image patch and passed separately to the black-box MLLM. This approach, while maximizing local detail preservation, inherently disrupts the global spatial context and inhibits inter-region integration. When the number of RoIs increases, the model is forced to reason over fragmented visual inputs without spatial continuity, which may lead to failures in tasks requiring holistic scene understanding or relational reasoning.

**Failure case: missed small object in ROI extraction.** Despite the overall gains, Zoomer may fail when the ROI extractor misses a small but question-critical object. Figure 6 shows an example where the question asks: *"Is the dog on the left or right side of the golden tower?"* The dog is a tiny object in the scene. If the detector fails to propose an ROI covering the dog, Zoomer-Local will not include the dog in the reconstructed input, causing the MLLM to answer incorrectly. Zoomer-Adaptive may partially mitigate this issue by appending a global view, but it still cannot recover the missing evidence when the dog becomes indistinguishable in the down-scaled overview. This highlights a key limitation of detector-guided visual focus and motivates future detector-free visual focusing strategies.

These findings suggest that preserving global structure—even at the cost of slightly reduced local detail—is beneficial in tasks that involve complex spatial or semantic dependencies. They also underscore the need for future designs to better balance granularity with contextual coherence in visual prompting pipelines.

### 5.4  Ablation Study

To assess the contribution of individual components in Zoomer, we conducted a comprehensive ablation study along two primary dimensions: (1) the role of multi-scale visual emphasis strategies, and (2) the impact of different RoI localization models. The results are summarized in Table 5.

| Method | CVBench-2D | CVBench-3D | VSTAR |
|---|---|---|---|
| Zoomer-Patches | 0.746 | 0.858 | 0.717 |
| Random-Patches | 0.571(-0.175) | 0.613(-0.245) | 0.597(-0.120) |
| Uniform-Patches | 0.637(-0.109) | 0.731(-0.127) | 0.664(-0.047) |

Table 8: Comparison of patch-selection strategies under the same patch budget. All methods use the same number of patches; Zoomer-Patches gains come from semantic selection and spatial composition.

**Patch selection under a fixed budget.** As a stronger baseline than Raw and Resize, we compare Zoomer-Patches with Random and Uniform patch selection under the same patch budget (Table 8). Since all three strategies transmit the same number of patches, the improvements of Zoomer-Patches can only come from semantically guided region selection and spatial-preserving composition, rather than from simply increasing visual resolution.

**Visual Emphasis Strategies.** We compare three emphasis methods: (i) a baseline strategy using the full image without emphasis ("Default"), (ii) a multi-resolution method that applies image resizing at various scales without cropping, and (iii) our proposed multi-scale strategy, which extracts overlapping RoI patches across hierarchical granularities.

Across all tested datasets, the multi-scale approach consistently outperforms the other two. For instance, on the *Vstar* and *CVBench* datasets, the multi-scale strategy yields relative improvements of 8.2% and 6.4% over the multi-resolution method, respectively. This suggests that the hierarchical region recall provided by the multi-scale cropping mechanism compensates for potential object boundary fragmentation. In contrast, multi-resolution methods—though preserving global structure—negatively impact performance, likely due to the model's reliance on fixed-size input distributions during pretraining. Adjusting resolution without architectural adaptation may lead to degraded feature extraction in the frozen backbone of black-box MLLMs.

**Backbone Variants for Region Extraction.** We also evaluate the effect of using different visual backbone models for region proposal and saliency detection. Specifically, we compare GroundingDINO (Liu et al., b) and EVF-SAM as RoI extractors in the multi-scale visual emphasis pipeline. Both variants yield significant improvements over the baseline; however, GroundingDINO exhibits slightly higher accuracy across datasets. This result suggests that while both models effectively localize task-relevant regions, the localization precision and language grounding of GroundingDINO may offer a marginal advantage in aligning visual slices with text prompts.

**Area Ratio Threshold** $T_A$ We vary the threshold in Eq. equation 1 among $\{0.25, 0.35, 0.45\}$ and report results on the VSTAR validation set.

| $T_A$ | Accuracy | Relative token usage |
|---|---|---|
| 0.25 | 0.645 | 0.82 |
| 0.35 | 0.652 | 0.79 |
| 0.45 | 0.647 | 0.76 |

Table 9: Ablation on the area ratio threshold $T_A$ for Zoomer-Adaptive on VSTAR (val).

We observe that $T_A = 0.35$ achieves the best trade-off between accuracy and token efficiency: smaller thresholds append the global view too often, increasing token usage with marginal accuracy gains, while larger thresholds skip the global view in cases that benefit from additional context.

**Synergistic Effects.** Combining the multi-scale emphasis with the Patches variant of Zoomer yields the strongest overall performance, validating the importance of both region-level granularity and architectural alignment. These findings highlight that visual prompting should be co-designed with content-aware preprocessing and prompt-space organization, especially under token-constrained inference regimes.

# 6 Conclusion

This paper presents Zoomer, a novel visual prompting framework that addresses the challenge of preserving fine-grained visual detail under token constraints in black-box multimodal language models (MLLMs). By decomposing the prompting task into prompt-aware emphasis, spatially structured orchestration, and budget-aware input construction, Zoomer enables effective visual grounding without access to model internals or architectural modifications.

Comprehensive evaluations across datasets such as *Vstar*, *RealWorldQA*, and *TerraIncognita* demonstrate that Zoomer consistently improves task accuracy while reducing token consumption. For example, Zoomer-Patches achieves a 26.9% gain over baseline accuracy on *Vstar*, while Zoomer-Adaptive improves *RealWorldQA* accuracy by 12.1%. On *TerraIncognita*, Zoomer achieves higher accuracy with 67% fewer tokens, highlighting its practical utility in resource-constrained settings.

While the present work focuses on token-level efficiency, an important emerging challenge is the communication cost associated with transferring high-resolution images from edge devices to cloud-based MLLMs. This concern is particularly relevant in scenarios such as wearable computing or mobile robotics. A promising direction for future research is to extend Zoomer toward edge-aware visual prompting, enabling lightweight pre-processing directly on-device to reduce upstream bandwidth and latency while maintaining task performance.

In summary, Zoomer contributes a modular and model-agnostic framework for enhancing visual processing in black-box MLLMs. Its integration of spatial structure, semantic relevance, and token budget constraints offers a principled foundation for both immediate deployment and future extensions in bandwidth-constrained or real-time vision-language applications.

**From heuristics to explicit optimization.** The current Zoomer pipeline is heuristic in nature: it relies on prompt-aware detection, spatial-preserving composition, and simple token budgeting rules. A natural next step is to formalize adaptive visual token allocation as a constrained optimization problem, e.g., maximizing task accuracy or expected utility under a fixed visual-token budget and latency constraint, and to develop trainable surrogates or reinforcement learning policies that optimize this objective end-to-end.

**Beyond external detectors.** Zoomer currently relies on open-vocabulary detectors (e.g., GroundingDINO, EVF-SAM) applied in a multi-scale fashion, which can become a bottleneck when detector performance degrades. Our experiments partially mitigate this via Zoomer-Adaptive and Zoomer-Global, which automatically include a global view when the cropped area ratio is small. Nonetheless, exploring detector-light or detector-free mechanisms—such as leveraging internal attention maps or saliency predictions from open-source MLLMs—is an exciting direction for future work.

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
