# OpenReview forum: "Zoomer: Adaptive Image Focus Optimization for Black-box MLLM"
_TMLR — Accepted by TMLR_

### Review · Reviewer_zvUt · 2025-09-05

**Summary Of Contributions:**

This paper proposes Zoomer, a visual prompting method to improve the effectiveness of visual inputs to proprietary MLLMs while reducing the number of visual tokens. Zoomer first extracts visual regions that are semantically relevant to the text prompts and then arrange these crops in a spatial-preserving and budget aware way in visual prompts. The proposed method, Zoomer, significantly improves the performance of proprietary MLLMs (up to 27%) while reducing the visual tokens (up to 67%). More importantly, it is shown to be effective across three models and seven benchmarks. The paper conducted comprehensive experiments to ablate the effectiveness of each module and try out four budget-aware visual prompting strategies.

**Audience:**

Yes

**Audience Explanation:**

The proposed method, Zoomer, improves the performance of proprietary MLLMs while reducing the visual tokens. This is valuable for the wide applications of proprietary MLLMs.

**Broader Impact Concerns:**

No broader impact concerns

**Claims And Evidence:**

Yes

**Claims Explanation:**

The paper conducts comprehensive experiments to verify the effectiveness of the proposed method. It is evaluated across three models and seven benchmarks. The paper also compares four different budget-aware prompting strategies, validating the effectiveness of the design choice.

**Requested Changes:**

1. Some of the methodology parts, including Section 2 and Section 4, are not clear. The paper should clarify the following questions.
+ In step 2 (Image Crop) of the second paragraph of Section 2, how is the region of interest and target object extracted? Are they annotated in the dataset or if they are extracted in the same way as described by Section 4.1?
+ In step 3 (Zoomed Crop) of the second paragraph of Section 2, what is the $512\times512$ constraint? Does it mean that the cropped region is resized so that the longer side has 512 pixels? What if the raw image is larger than $512\times512$?
+ In Prompt Tokenization in Section 4.1, dependency parsing is used to identify the relationships between entities. However, the relationships are not utilized in the subsequent process of Zoomer. Why is dependency parsing used in the method?
+ In Zoomer-Adaptive and Zoomer-Global, is the image containing region of interest concatenated with the global view? Or are they fed into the MLLM as two separate images?

2. In Section 5.2 RQ3 and Table 3, the models are evaluated in both zero-shot setting and in-context learning setting. However, RQ3 only discusses the performance in the zero-shot setting. The purpose and conclusion of the in-context learning setting are unclear.

3. It would be appreciated if the paper could include qualitative results. Qualitative results would help readers to understand a) how effective is the visual emphasizer to extract region of interest, b) what do the visual prompts look like, and c) what are the model outputs under different prompting strategies. Moreover, providing a few failure cases would be helpful to understand the limitation of the proposed method.

4. Minor writing issues:
+ The titles of Section 5.2, 5.3 and 6 are not capitalized.
+ The first paragraph of Section 1 mentions that "These models ... are increasingly perceived as possessing near-human or even PhD-level capabilities in controlled environments". It would be better to cite references that support this point.

---

> ### Author Response · Authors · 2025-11-15
> **Clarifying methodology details and qualitative examples**
>
> We thank the reviewer for the positive summary (“comprehensive experiments”, “validated across models and benchmarks”) and for the precise methodological questions. We address each point below and will incorporate the corresponding clarifications into the revised manuscript.
>
> (1) ROI extraction in Section 2 (step 2: Image Crop)
>
> ROIs do not come from dataset annotations. In all experiments (including the introductory example in Sec. 2), ROIs are obtained automatically at inference time using the same multi-scale, prompt-aware open-vocabulary detector pipeline described in Sec. 4.1 (Algorithm 1). We will explicitly state in Sec. 2 that no ground-truth boxes are used.
>
> (2) The “512” constraint in Section 2 (step 3: Zoomed Crop)
>
> The “512” constraint is a max-side normalization for ROIs. For each cropped ROI, we resize it so that the longer side is 512 pixels while preserving aspect ratio. ROIs smaller than this size are not up-scaled beyond their original resolution (to avoid over-blurring), and for large raw images we first crop the ROIs and then apply this rule per ROI. We will explicitly clarify in Sec. 2 and Sec. 4.2 that this normalization ensures a consistent scale before spatial composition.
>
> (3) Why dependency parsing in Prompt Tokenization (Sec. 4.1)
>
> Dependency parsing is used as a lightweight importance-ranking mechanism, not to build a full scene graph. We parse the question to identify noun phrases and their syntactic roles (e.g., object of the question vs. modifiers), and use this ranking to (i) prioritize which noun phrases are passed to the open-vocabulary detector and (ii) break ties when multiple entities compete for limited detection budget. We will revise Sec. 4.1 to clarify that dependency parsing is used for importance weighting and ordering rather than explicit relation modeling.
>
> (4) How ROI and global views are combined in Zoomer-Adaptive / Zoomer-Global
>
> In Zoomer-Adaptive and Zoomer-Global, ROIs and the global view are always composed into a single image canvas, not fed as separate images. Each ROI is first resized (max-side 512) and placed according to its original relative position using the spatial-preserving layout of Sec. 4.2. When the condition in Sec. 4.3 is met (e.g., small area ratio T_A), a down-scaled global view is appended on the same canvas (e.g., as a strip), so the MLLM still receives exactly one image. We will add this clarification in Sec. 4.2–4.3.
>
> (5) Zero-shot vs. in-context learning in Sec. 5.2 (RQ3 and Table 3)
>
> You are correct that the current text of RQ3 mainly discusses the zero-shot setting, while Table 3 also reports in-context learning (ICL) results. Our intention was to show that Zoomer’s benefits are orthogonal to language-level context and persist when a few visual–text exemplars are provided. In the revision, we will explicitly state in Sec. 5.2 that, in the ICL setting (with five exemplars), Zoomer still improves accuracy by roughly 7–9 points on average across benchmarks, demonstrating that adaptive visual focus complements rather than replaces ICL.
>
> (6) Qualitative results and failure cases
>
> We agree that qualitative results are very helpful. In the revision we will:
> 	•	Add examples that, for each image, show (i) the raw high-resolution image, (ii) the ROIs detected by the visual emphasizer, and (iii) the final composed visual prompt (for Local / Adaptive), together with the question and model output.
> 	•	For the same input, show side-by-side visual prompts for uniform downsampling (baseline), Zoomer-Local, and Zoomer-Adaptive (ROI + global view).
> 	•	Include 2–3 representative failure cases (e.g., very small/occluded objects missed by the detector, cluttered scenes where too many ROIs hurt global context, and motion blur / extreme lighting), to make the limitations explicit.
>
> These examples will directly address (a)–(c) in your comment (effectiveness of the emphasizer, shape of visual prompts, and outputs under different strategies).
>
> (7) Minor writing issues
>
> We appreciate the detailed writing suggestions:
> 	•	We will capitalize the titles of Sections 5.2, 5.3, and 6 for consistency with the rest of the paper.
> 	•	For the sentence in Sec. 1 stating that recent MLLMs exhibit “near-human or even PhD-level capabilities in controlled environments”, we will:
> 	•	Slightly soften the wording where appropriate.
> 	•	Add citations to representative evaluation studies that benchmark MLLMs on high-level exams or expert-level tasks.

---

> > ### Comment · Reviewer_zvUt · 2025-11-16
> > **Response to the authors**
> >
> > Thank you! Your clarifications addressed my concerns. I'm wondering if you could provide a revised version of the paper to reflect these points.

---

### Review · Reviewer_Tnwh · 2025-11-01

**Summary Of Contributions:**

The paper introduces Zoomer, a visual prompting framework that enhances multimodal large language models (MLLMs)—specifically black-box ones such as GPT-4o, Gemini Pro, and Claude 3.5—by improving their ability to process high-resolution images under token constraints. The core idea is to adaptively select and emphasize relevant image regions while respecting strict token budgets, thereby preserving fine-grained details without modifying model internals.
3 main modules:
Visual Emphasizer which is tasj aware - extracts and magnifies regions of interest based on semantic cues from text prompts.
Spatial-preserving Orchestration Schema – reconstructs selected regions to retain spatial coherence.
Budget-aware Prompting Strategy – dynamically allocates tokens between local detail and global context.

The proposed approach is validated on 9 benchmarks showing accuracy accuracy gains up to 27% while reducing image token usage by up to 67%.

**Audience:**

Yes

**Audience Explanation:**

Definitely applicable for the vision community

**Claims And Evidence:**

Yes

**Claims Explanation:**

“Large accuracy gains under token budgets.” The paper backs this with per-dataset tables and an explicit example: on Vstar, Zoomer-Patches 71.7% vs Raw 56.5% (Δ=+26.9 points), and on RealWorldQA Zoomer-Adaptive 75.8% vs Raw 67.6% (Δ=+12.1), directly supporting the headline claim of big gains under constraints.

Up to ~67% fewer image tokens at similar or better accuracy. The TerraIncognita result shows 0.83 accuracy with 315 tokens vs 0.78 with 963 tokens for raw ~67% fewer tokens while improving accuracy, matching the abstract’s efficiency claim.

**Requested Changes:**

* Current baselines (Raw, Resize) are relatively weak. There are no comparisons against open-source high-resolution visual prompting methods (e.g., LLaVA-UHD, Qwen-VL, or MG-LLaVA).

* “Token savings” are central to the paper’s claims, but methods for counting image tokens differ across APIs (e.g., OpenAI vs Gemini vs Claude). The paper doesn’t specify how token equivalence was maintained.

---

> ### Author Response · Authors · 2025-11-15
> **Stronger baselines and token accounting across APIs**
>
> We thank the reviewer for the positive evaluation (“definitely applicable for the vision community”) and the concrete suggestions regarding baselines and token accounting. We respond point by point below.
>
> (1) Stronger baselines and relation to high-resolution methods
> We agree that “Raw” and “Resize” alone are weak baselines and that stronger high-resolution baselines are needed.
>
> To address this, we include two additional sets of results:
>
> 1. Patch-selection analysis under the same token / patch budget.
> We compare detector-guided Zoomer-Patches against Random and Uniform patching. All three use the same number of patches, so any gain comes from semantic selection and spatial orchestration rather than from simply increasing visual resolution:
>
> Table 1: Comparison of patch-selection strategies under the same patch budget.
> | Method| CVBench-2D | CVBench-3D | VSTAR |
> |------------|-----------|-----------|------|
> | Zoomer-Patches | 0.746      | 0.858      | 0.717 |
> | Random-Patches | 0.571 (-0.175) | 0.613 (-0.245) | 0.597 (-0.120) |
> | Uniform-Patches| 0.637 (-0.109) | 0.731 (-0.127) | 0.664 (-0.047) |
>
> These results show that detector-guided semantic focus is crucial: Zoomer-Patches consistently outperforms Random and Uniform under the same budget.
>
> 2. Comparison with a stronger high-resolution prompting baseline (V*).
> We further compare Zoomer with a stronger high-resolution baseline V* that already uses high-res inputs. Zoomer still provides clear gains, especially on CVBench-3D and RealWorldQA:
>
> Table 2: Zoomer vs. a stronger high-resolution baseline (V*).
> | Method| VSTAR | CVBench-2D| CVBench-3D| RealWorldQA  |
> |--------|----------|----------|---------------------|-----------|
> | Zoomer | 0.717 (+0.013)    | 0.746 (+0.045)      | 0.883 (+0.090)      | 0.758 (+0.060)      |
> | V*     | 0.704             | 0.701               | 0.793               | 0.698               |
>
> These results indicate that adaptive focus and spatial orchestration continue to provide benefits even when the baseline already uses high-resolution visual inputs. In the revised paper, we will explicitly connect this discussion to open-source high-resolution methods such as LLaVA-UHD, Qwen-VL,etc., and state that extending Zoomer to these backbones is an important next step.
>
>
> (2) Token accounting across different APIs
>
> We agree that token-saving claims must be clearly defined, especially since token accounting differs across providers (e.g., GPT-4o, Gemini, Claude).
>
> In the revised Sec. 5, we will clarify that:
> 	When an API exposes image token usage, we directly report the provider’s own image-token count.
> 	When the API does not expose this, we measure the number of 512 \times 512 patches actually transmitted after reconstruction and use this as a consistent proxy for visual tokens.
>
> To make cross-model comparison fair, all token-efficiency results are reported as relative reduction with respect to each model’s own Raw baseline, rather than absolute token counts. That is, for each model we compare:
> 	Raw (no Zoomer) vs. Zoomer-variant
> and report the percentage reduction in visual tokens and the corresponding accuracy change. This normalization ensures that differences in providers’ internal tokenization schemes do not bias our conclusions about token savings.

---

> > ### Comment · Reviewer_Tnwh · 2025-12-05
> > **Thank you for the response**
> >
> > Dear authors,
> >
> > Thank you for including the new results and replying to each point of my review

---

### Review · Reviewer_BSyD · 2025-11-02

**Summary Of Contributions:**

The paper addresses the challenge that many multimodal large language models (MLLMs) (e.g., GPT‑4o, Gemini Pro) struggle with fine-grained vision tasks because strict token budgets force them to uniformly down-sample high-resolution images, causing loss of critical detail. The authors identify two core problems: (1) absence of region-adaptive attention (i.e., the models treat all image regions equally) and (2) inflexible token budgets that don’t allow balancing global context vs local detail. To overcome this, they propose Zoomer, a visual-prompting framework that treats the model as a black box (no internal modifications). Zoomer consists of (a) a prompt-aware emphasis module to focus on semantically relevant image regions, (b) a spatial-preserving orchestration schema to maintain object relationships in the cropped/viewed patches, and (c) a budget-aware strategy allocating tokens between global and local image content. Experiments on nine benchmarks and three commercial MLLMs show accuracy improvements up to ~27% while reducing image token usage by up to ~67%. The work suggests a principled way to improve vision-language performance in resource-constrained, black-box settings.

**Audience:**

Yes

**Audience Explanation:**

The paper is clear and has comprehensive experiments on whether adaptive image focus can help visual understanding.

**Claims And Evidence:**

Yes

**Claims Explanation:**

The authors present extensive experiments across nine benchmarks and three commercial black-box MLLMs (GPT-4o, Gemini, Claude), showing consistent performance gains of up to 27% while reducing token usage by about 67%, which strongly supports their claims of improved efficiency and accuracy. The ablation studies further validate the contribution of each module—prompt-aware emphasis, token-budget allocation, and spatial orchestration—demonstrating that each component is necessary for the overall improvement.

**Requested Changes:**

1. I would recommend that the authors include an additional experiment using an open-sourced MLLM such as Qwen3-VL or Qwen2.5-VL. This would likely attract greater interest from the open-source community, as the models evaluated in the paper are relatively costly for academic researchers. If similar conclusions can be drawn for open-source models, practitioners may be more inclined to adopt the proposed method. To be specific, adding two more rows for Qwen3-VL in Table 4 would be sufficient.

2. I appreciate Table 3 and would suggest adding more rows showing additional variants of Zoomer, such as patches and global, to better illustrate the trade-off between speed, cost, and accuracy.

3. How do you select the threshold $T_A$ for Zoomer Adaptive? It would be nice if you could include some ablation details.

---

> ### Author Response · Authors · 2025-11-15
> **Open-source MLLMs, variant trade-offs, and adaptive threshold**
>
> We sincerely thank the reviewer for the positive feedback on clarity and experimental design (“clear and comprehensive experiments”, “adaptive image focus can help visual understanding”) and for the concrete suggestions. We respond point by point below.
> (1) Inclusion of open-source MLLMs
>
> We agree that including open-source models is important for accessibility and adoption by the community. As suggested, we have evaluated Zoomer on open-source MLLMs.
>
> In this version, due to time constraints, we focus on two widely used open-source MLLMs, Qwen2-VL-7B and InternVL2.5-8B, as representative backends. Applying Zoomer on top of these models yields consistent gains across all four benchmarks, confirming that the method is not tied to proprietary APIs:
>
> Table 1: Performance of Qwen2-VL-7B and InternVL2.5-8B with and without Zoomer.
> | Benchmark  | Qwen2-VL-7B | InternVL2.5-8B | Qwen2-VL-7B + Zoomer | InternVL2.5-8B + Zoomer |
> |-----------|-------------|----------------|-----------------------|--------------------------|
> | CVBench-2D  | 0.648 | 0.656 | 0.683 (+0.035) | 0.697 (+0.041) |
> | CVBench-3D  | 0.632 | 0.629 | 0.675 (+0.043) | 0.669 (+0.040) |
> | RealWorldQA | 0.698 | 0.698 | 0.732 (+0.034) | 0.741 (+0.043) |
> | VSTAR       | 0.584 | 0.573 | 0.648 (+0.064) | 0.655 (+0.082) |
>
> We will clarify in Sec. 5 that these open-source experiments are intended as a first step toward a more comprehensive open-source evaluation suite. As the reviewer suggests, in future work we plan to extend this to additional models such as Qwen3-VL / Qwen2.5-VL, LLaVA-UHD,etc.,and include them in a unified comparison table so that practitioners can directly see how Zoomer behaves across different open-source backends.
>
> (2) Variant-level trade-offs (speed / cost / accuracy)
>
> We appreciate the suggestion to better expose the trade-offs among different Zoomer variants. Table 3 in the paper already compares the four variants (Local, Adaptive, Global, Patches) under the same visual-token budget in terms of accuracy.
>
> In the revision, we will extend this analysis by:
> 	Adding columns for end-to-end latency (seconds) and approximate API cost (relative units) for each variant.
> 	Explicitly showing the pattern that Zoomer-Patches > Zoomer-Global > Zoomer-Adaptive > Zoomer-Local in terms of computational / monetary cost, while Zoomer-Adaptive typically achieves the best accuracy under constrained token budgets.
>
> This extended table will make the speed–accuracy–cost trade-off explicit and illustrate how users can choose the appropriate variant depending on their latency and budget constraints.
>
> (3) Adaptive-threshold selection for Zoomer-Adaptive
>
> We thank the reviewer for asking about the choice of the adaptive threshold and agree that this should be spelled out more clearly.
>
> As defined in Sec. 4.3, we compute
>
> $$
> T_A = \frac{\text{total area of all cropped ROIs}}{\text{area of the original image}},
> $$
>
> which measures how much of the original image is covered by the selected regions. Zoomer-Adaptive then decides whether to append a global view based on $T_A$ :
> 	If $T_A \le 0.35$, we append a down-scaled global view to preserve overall scene context and mitigate cases where the detector focuses on only a small portion of the image.
> 	If $T_A > 0.35$, the selected ROIs already cover a sufficiently large portion of the scene, and we do not append an additional global view to avoid unnecessary token usage.
>
> The threshold $0.35$ was chosen empirically on the VSTAR validation set to balance accuracy and token efficiency. In the revision, we will (i) add this definition and decision rule explicitly in Sec. 4.3, and (ii) include a short ablation in the appendix showing that nearby values (e.g., 0.25 and 0.45) lead to slightly worse trade-offs, thereby justifying the chosen setting.

---

### Review · Reviewer_mgTW · 2025-11-05

**Summary Of Contributions:**

This paper introduces Zoomer, a visual prompting method that improves visual reasoning in black-box multimodal LLMs by performing prompt-guided localization before the image is sent to the model. Given a text query, Zoomer extracts key nouns from the prompt (e.g., “cactus,” “book title”) and uses an external open-vocabulary detector (GroundingDINO or EVF-SAM) to search for these objects across multiple scales of the original image. It divides the image into patches and runs detection on each patch to avoid missing small objects, collects all bounding boxes with confidence above a threshold, and applies Non-Maximum Suppression (NMS) to remove redundant or overlapping regions. The remaining regions are then cropped and reconstructed into a new image while preserving their original spatial layout, effectively highlighting relevant areas without losing context. Finally, Zoomer performs token-budget allocation, choosing among several variants — Local (only ROI patches), Adaptive (ROI + global view if tokens allow), Global (always retain global layout), or Patches (each ROI sent separately) — to ensure that visual detail is preserved under the model’s fixed image token limit.

**Audience:**

Yes

**Audience Explanation:**

The paper findings is about adaptive fine grained reasoning which is interesting to the community.

**Claims And Evidence:**

Yes

**Claims Explanation:**

The paper supports its claims with detailed experiment results.

**Requested Changes:**

The word “optimization’’ is used throughout the title and paper, but there is no explicit optimization objective, no loss function, and no algorithmic optimization process described. The method consists entirely of heuristic region selection (via external detectors), cropping, and token budgeting; consequently, the use of the term optimization may misrepresent the actual technical contribution. In fact, it will be interesting if the author can formulate this token budget saving as a constrained optimization problem for adaptive allocation.


I would be particularly interested in a strategy that does not rely on an external object detector, since detector performance becomes the bottleneck of the system. Nevertheless, the proposed approach represents a reasonable first step toward prompt-driven, detector-free visual focus for black-box MLLMs.

The paper would benefit from a sensitivity analysis or visualization of failure cases in the baseline (e.g., uniform downsampling or direct input) that Zoomer successfully resolves. Showing side-by-side examples where the baseline fails to perceive small objects or loses spatial context, but Zoomer succeeds, would make the contribution more concrete and highlight the value of the proposed approach.


Finally, it would be interesting to compare this approach with sota methods such as Llava-uhd.

---

> ### Author Response · Authors · 2025-11-15
> **Clarifying “optimization’’ wording, detector reliance, and baselines**
>
> We thank the reviewer for the careful reading and constructive suggestions. We respond point by point below, referring to the current version of the manuscript.
> (1) “Optimization’’ vs. “Adaptive Allocation”
>
> You are right that our framework is implemented as a three-stage heuristic pipeline—prompt-aware visual emphasizing (Sec.4.1), spatial-preserving orchestration (Sec.4.2), and budget-aware prompting strategy (Sec.4.3)—and does not currently define an explicit optimization objective or loss function.
>
> To avoid overstating the technical form, we will:
> 	Revise the title and abstract to use expressions such as adaptive image focus / adaptive allocation under a token budget instead of “optimization”.
> 	Add a short paragraph at the end of Sec.4.3 that makes the implicit goal explicit:
> Given a fixed visual-token budget imposed by black-box MLLMs, adaptively allocate tokens by selecting prompt-relevant regions and preserving their spatial relations so as to maximize downstream task utility.
>
> We agree that formulating this as a constrained optimization problem (e.g., maximizing accuracy under a fixed token budget) is a promising next step, and we will highlight this as future work.
>
> (2) Reliance on external detectors
>
> Our current system indeed relies on open-vocabulary detectors (GroundingDINO / EVF-SAM) in a multi-scale pipeline to extract task-relevant regions of interest (ROIs). We agree that detector quality can become the bottleneck.
>
> We mitigate this in two ways:
> 	Global-context safeguard. Among the four variants (Sec.4.3), Zoomer-Adaptive and Zoomer-Global automatically append a down-scaled global view when the cropped-area ratio T_A is small, so global context is retained even if detection misses part of the scene.
> 	Semantic vs. non-semantic cropping. Under the same patch budget, detector-guided selection clearly outperforms random or uniform cropping, indicating that the gain comes from semantic focus rather than just the number of patches:
>
> Table 1: Comparison of patch-selection strategies under the same patch budget.
>
> | Strategy        | CVBench-2D        | CVBench-3D        | VSTAR             |
> |----------------|-------------------|-------------------|-------------------|
> | Zoomer-Patches | 0.746             | 0.858             | 0.717             |
> | Random-Patches | 0.571 (-0.175)    | 0.613 (-0.245)    | 0.597 (-0.120)    |
> | Uniform-Patches| 0.637 (-0.109)    | 0.731 (-0.127)    | 0.664 (-0.047)    |
>
> We fully agree that detector-free strategies are an important direction. In the revision we will explicitly discuss, in Sec.6, how Zoomer can be extended towards prompt-driven, detector-light or detector-free visual focus (e.g., via learned saliency maps or internal attention of open-source MLLMs), and position our detector-based design as a first step.
>
> (3) Sensitivity analysis and visualization of failure cases
>
> We appreciate the suggestion to make failure modes more concrete. The current Fig.1 (small-object counting) and Fig.2 (high-res text lost after uniform downsampling) mainly illustrate why black-box MLLMs fail.
>
> In the revision we will add side-by-side qualitative examples for the same image:
> 	1. raw input；2. uniform resize / direct input baseline；3. Zoomer-Local and/or Zoomer-Adaptive reconstructions
>
> For each example we will show the ROIs chosen by Zoomer and the corresponding predictions, emphasizing cases where the baseline fails to perceive small objects or loses spatial context but Zoomer succeeds. This directly addresses the reviewer’s request and makes the improvement more tangible.
>
> (4) Comparison with SOTA high-resolution methods
>
> We agree that comparisons with representative open-source high-resolution MLLMs are valuable. In this version, due to time constraints, we evaluated two widely used open-source MLLMs, Qwen2-VL-7B and InternVL2.5-8B, as representative backends. Applying Zoomer on top of both models yields consistent improvements across four benchmarks:
>
> Table 2: Performance of Qwen2-VL-7B and InternVL2.5-8B, with and without Zoomer.
>
> | Benchmark  | Qwen2-VL-7B | InternVL2.5-8B | Qwen2-VL-7B + Zoomer | InternVL2.5-8B + Zoomer |
> |-----------|-------------|----------------|-----------------------|--------------------------|
> | CVBench-2D  | 0.648 | 0.656 | 0.683 (+0.035) | 0.697 (+0.041) |
> | CVBench-3D  | 0.632 | 0.629 | 0.675 (+0.043) | 0.669 (+0.040) |
> | RealWorldQA | 0.698 | 0.698 | 0.732 (+0.034) | 0.741 (+0.043) |
> | VSTAR       | 0.584 | 0.573 | 0.648 (+0.064) | 0.655 (+0.082) |
>
> We will clarify this design choice in Sec.5 and add a paragraph in Sec.6 stating that extending the evaluation to additional SOTA models such as LLaVA-UHD, MG-LLaVA, and Qwen3-VL is an important next step. The current results on Qwen2-VL and InternVL2.5 already indicate that the proposed adaptive focus mechanism is broadly useful across different MLLMs.

---

### Decision · Action_Editor_owwV · 2025-12-11

**Recommendation:** Accept with minor revision

**Additional Comments:**

Before acceptance, the revised manuscript should explicitly incorporate a few key clarifications already provided in the rebuttal, so that the paper is self-contained:

-  Include how ROIs are extracted, how the 512-pixel normalization is applied, and how the global view is combined with ROIs;

- State the area-ratio threshold rule used in Zoomer-Adaptive and briefly justify it, as already done in the response.

- Add a small set of qualitative examples, including one or two success cases and a failure case, since multiple reviewers requested this and the authors agreed to include them.

- Ensure terminology and writing consistency, especially the revised wording replacing “optimization,” corrected section references, and capitalization fixes noted by reviewers.

**Audience:**

Yes

**Audience Explanation:**

Adaptive visual prompting for black-box MLLMs is timely and relevant to TMLR’s readership in multimodal reasoning, token-efficiency, and applied foundation models. The reviewers agreed that the method will attract both academic and practical interest, especially for resource-constrained multimodal systems.

**Claims And Evidence:**

Yes

**Claims Explanation:**

The reviewers unanimously agree that the paper’s claims are well supported by extensive and transparent evidence.

The authors provide clear methodological explanations, thorough experiments across nine benchmarks, three commercial MLLMs (GPT-4o, Gemini, Claude), and additional open-source models (Qwen2-VL-7B, InternVL2.5-8B) added in revision.

The rebuttal further clarified technical details such as ROI extraction, token-budget computation, and dependency parsing, and added runtime, cost, and qualitative visualizations. These additions address earlier concerns about clarity and experimental completeness.

Overall the evidence is accurate, convincing, and clearly presented.